# Dalpiciclib and pyrotinib in women with HER2-positive advanced breast cancer: a single-arm phase II trial

Min Yan [1] ✉, Limin Niu[1], Huimin Lv[1], Mengwei Zhang[1], Jing Wang[1], Zhenzhen Liu [1], Xiuchun Chen[1], Zhenduo Lu[1], Chongjian Zhang[1], Huiai Zeng[1], Shengnan Zhao[1], Yajing Feng[1], Huihui Sun[1] & Huajun Li[2]

CDK4/6 inhibitors have shown a synergistic effect with anti-HER2 therapy in hormone receptor (HR)-positive and HER2-positive breast cancer (BC). In this phase 2 study (NCT04293276), we aim to evaluate a dual-oral regimen of CDK4/6 inhibitor dalpiciclib combined with HER2 tyrosine kinase inhibitor pyrotinib as front-line treatment in women with HER2-positive advanced BC ($n = 41$) including those with HR-negative disease. The primary endpoint is the objective response rate, and secondary endpoints include progression-free survival (PFS), overall survival (OS), and safety. With a median follow-up of 25.9 months, 70% (28/40) of assessable patients have a confirmed objective response, meeting the primary endpoint. The median PFS is 11.0 months (95% CI = 7.3–19.3), and OS data are not mature. The most common grade 3 or 4 treatment-related adverse events (AEs) are decreased white blood cell count (68.3%), decreased neutrophil count (65.9%), and diarrhea (22.0%). Most AEs are manageable, and no treatment-related deaths occur. These findings suggest that this combination may have promising activity and manageable toxicity. Further investigation is needed.

Cyclin D-cyclin-dependent kinase 4 and 6 (CDK4/6) axis is known to play a key role in the cell cycle and its dysregulation is one of the central mechanisms of breast cancer (BC) biology[1]. CDK4/6 inhibitors block the cell cycle by inhibiting the kinase activity of the CDK/cyclin complex and have been proven to be effective in hormone receptor (HR)-positive and human epidermal growth factor receptor 2 (HER2)-negative BC[2]. In terms of HER2-positive BC, CDK4/6 is an attractive target as well, as it lies downstream of the HER2 signal pathway, driving resistance to HER2-targeted therapies[3]. It has been found that HER2 positivity could induce significantly higher levels of CDK4/6 activity, suggesting that HER2-positive BC may respond to CDK4/6 inhibitors[4]. Meanwhile, preclinical studies have shown that CDK4/6 inhibitors have an inhibitory effect on resistant and non-resistant HER2-positive cell lines, and can delay the recurrence of HER2-driven BC[5]. In addition, CDK4/6 inhibitors were found to be able to sensitize patient-derived xenograft tumors to HER2-targeted therapies and showed synergistic effect when combined with HER2-targeted therapies[3,5]. Clinical trials have demonstrated the efficacy of the combination of CDK4/6 inhibitors and HER2-targeted agents in HR-positive and HER2-positive advanced BC (ABC)[6,7]. However, whether the benefit of the combination can expand to all HER2-positive BCs requires further exploration.

Dalpiciclib (SHR6390) is a highly selective, small molecule CDK4/6 inhibitor, recently approved for HR-positive, HER2-negative ABC in China, based on impressive efficacy and manageable toxicity in a randomized phase 3 DAWNA-1 trial[8]. Pyrotinib is an oral, irreversible pan-ErbB receptor tyrosine kinase inhibitor (TKI) targeting HER1, HER2, and HER4, which has been proven effective in the PHEOBE[9] and PHENIX[10] trials, and approved for the treatment of HER2-positive ABC in China[11]. Notably, it has been reported that the combination of the

[1]Department of Breast Disease, Henan Breast Cancer Center/The Affiliated Cancer Hospital of Zhengzhou University & Henan Cancer Hospital, Zhengzhou, China. [2]Jiangsu Hengrui Pharmaceuticals Co., Ltd, Shanghai, China. ✉e-mail: ym200678@126.com

two drugs could enhance the anti-tumor effect in HER2-positive and HR-positive BC cell lines and xenograft models[12].

Our study aims to investigate the dual-oral regimen of dalpiciclib and pyrotinib, a CDK4/6 inhibitor and a HER2 TKI without a backbone of chemotherapy or endocrine therapy, in HER2-positive ABC patients irrespective of HR status. Here we report the results.

## Results

### Patient allocation and baseline characteristics

Between April 9, 2020, and May 19, 2021, a total of 42 patients were screened and 41 were enrolled and received study treatment. As per protocol, 24 patients were enrolled in the first stage and 17 responses were achieved, reaching the prespecified target for this stage, so the recruitment continued. One patient's updated information showed that she had a HER2 immunohistochemistry (IHC) score of 1+ and HER2 gene amplification by fluorescence in-situ hybridization (FISH), which did not meet the inclusion criteria. Thus, the patient was excluded from the efficacy analysis though she received study treatment (Fig. 1). Baseline characteristics of the participants are summarized in Table 1.

### Efficacy

By the data cutoff date on May 9, 2023, the median follow-up for the 40 response-evaluable patients was 25.9 months (interquartile range [IQR] = 22.7–30.0). Twenty-eight had a confirmed objective response (two achieved complete response and 26 achieved partial response), with an objective response rate (ORR) of 70% (95% CI = 53.5–83.4%; Fig. 2).

The median progression-free survival (PFS) was 11.0 months (95% CI = 7.3–19.3), with an estimated 12-month PFS rate of 44.7% (95% CI = 28.5%-59.7%; Fig. 3). There were ten deaths recorded after disease progression, and overall survival (OS) data were not

mature. The estimated 12-month and 18-month OS rates were 90.0% (95% CI 75.5%-96.1%) and 82.5% (95% CI 66.8%-91.2%), respectively.

**Table 1 | Demographic and clinical characteristics at baseline**

| | Total (n = 41) | HR + (n = 18) | HR- (n = 23) |
|---|---|---|---|
| Age | | | |
| Median (range), years | 53.0 (28–68) | 52.5 (30–65) | 53.0 (28–68) |
| ECOG performance status, n (%) | | | |
| 0 | 1 (2.4) | 0 (0) | 1 (4.3) |
| 1 | 40 (97.6) | 18 (100) | 22 (95.7) |
| Hormone receptor status, n (%) | | | |
| HR+ | 18 (43.9) | 18 (100) | — |
| ER+ or PR+ | 11 (26.8) | 11 (61.1) | — |
| ER+ and PR+ | 7 (17.1) | 7 (38.9) | — |
| HR-[a] | 23 (56.1) | — | 23 (100) |
| HER2 status, n (%) | | | |
| IHC 3+ | 30 (73.2) | 11 (61.1) | 19 (82.6) |
| IHC 2+ and FISH + | 10 (24.4) | 7 (38.9) | 3 (13.0) |
| IHC 1+ and FISH + | 1 (2.4)[b] | 0 (0) | 1 (4.3) |
| Metastatic sites at screening, n (%) | | | |
| Visceral | 38 (92.7) | 16 (88.9) | 22 (95.7) |
| Brain | 13 (31.7) | 5 (27.8) | 8 (34.8) |
| Lung | 23 (56.1) | 9 (50.0) | 14 (60.9) |
| Liver | 17 (41.5) | 10 (55.6) | 7 (30.4) |
| Non-visceral | 3 (7.3) | 2 (11.1) | 1 (4.3) |
| No. of prior systemic treatments, n (%) | | | |
| 0 | 3 (7.3) | 2 (11.1) | 1 (4.3) |
| 1 | 27 (65.9) | 10 (55.6) | 17 (73.9) |
| 2 | 11 (26.8) | 6 (33.3) | 5 (21.7) |
| Median (range) | 1 (0–2) | 1 (0–2) | 1 (0–2) |
| Prior systemic treatment, n (%) | | | |
| None | 3 (7.3) | 2 (11.1) | 1 (4.3) |
| Neoadjuvant/adjuvant | 29 (70.7) | 13 (72.2) | 16 (69.6) |
| Metastatic/locally advanced | 21 (51.2) | 10 (55.6) | 11 (47.8) |
| Previous HER2-target therapy, n (%) | | | |
| Trastuzumab | 28 (68.3) | 10 (55.6) | 18 (78.3) |
| Pertuzumab | 5 (12.2) | 0 (0) | 5 (21.7) |
| None | 13 (31.7) | 8 (44.4) | 5 (21.7) |
| Resistance to trastuzumab[c], n (%) | | | |
| Yes | 16 (39.0) | 5 (27.8) | 11 (47.8) |
| No | 25 (61.0) | 13 (72.2) | 12 (52.2) |
| Previous chemotherapy, n (%) | | | |
| Yes | 38 (92.7) | 16 (88.9) | 22 (95.7) |
| No | 3 (7.3) | 2 (11.1) | 1 (4.3) |
| Previous endocrine therapy[d], n (%) | | | |
| Yes | 12 (66.7) | 12 (66.7) | — |
| No | 12 (66.7) | 6 (33.3) | — |

HR hormone receptor, ECOG Eastern Cooperative Oncology Group, ER estrogen receptor, PR progesterone receptor, HER2 human epidermal growth factor receptor 2, IHC immunohistochemistry, FISH fluorescence in-situ hybridization.
[a]HR- was defined as ER/PR expression in <1% tumor cells.
[b]The patient with HER2 IHC 1 + /FISH + had HR-negative disease and metastasis in the lung, liver, and lymph node, and was previously treated with chemotherapy combined with trastuzumab and pertuzumab as first-line regimen at the advanced setting and resistant to trastuzumab.
[c]Resistance to trastuzumab was defined as relapse during or within 12 months after adjuvant trastuzumab or progression within 6 months of trastuzumab treatment for metastatic disease.
[d]Data were calculated in 18 HR+ patients.

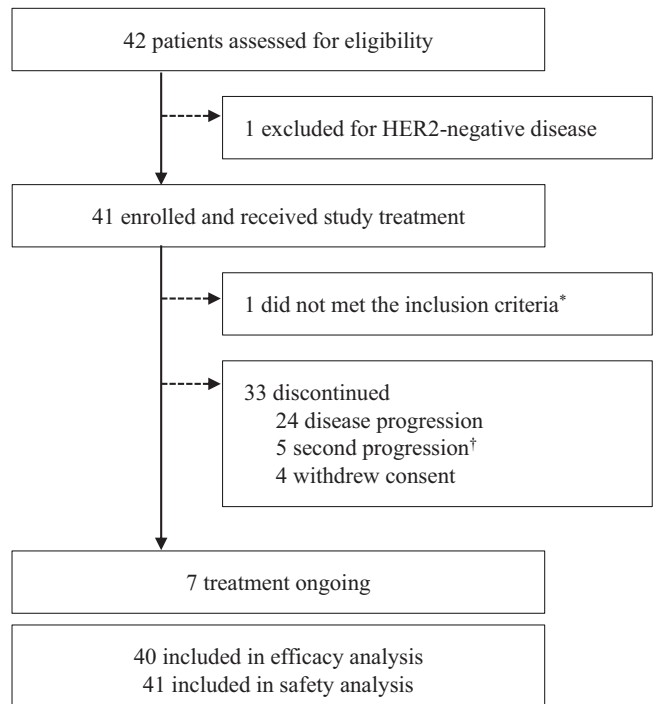

**Fig. 1 | Trial profile.** * One patient was found with HER2 IHC 1+ after receiving study treatment and was excluded from efficacy analysis due to not meeting the inclusion criteria. † Five patients with brain metastasis at baseline and only intracranial progression during the study period received local stereotactic radiosurgery and resumed study treatment until the second progression at the discretion of the patients and treating physicians.

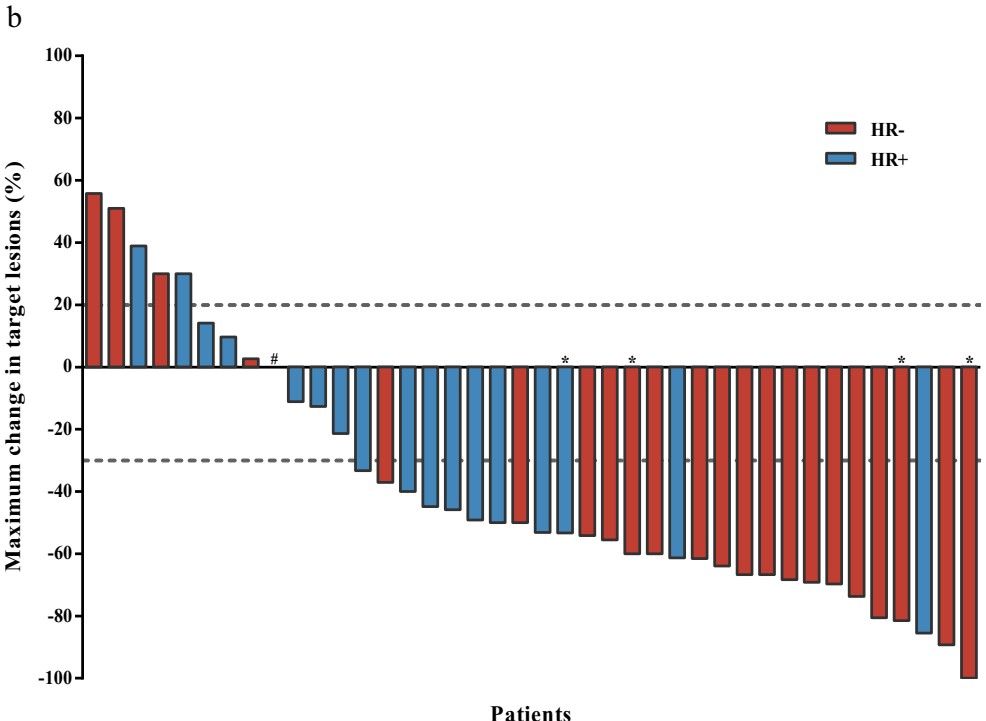

**Fig. 2 | Tumor response. a**. Swimmer's plot showing patterns of response in patients and the duration of response. **b.** Waterfall plot. All objective responses were confirmed by repeated imaging 4–6 weeks later. * Represents patients achieving complete response in target lesions, including three who only had lymph node target lesions. # One patient with HR-positive disease had no change in target lesions at the end of the treatment from baseline. HR, hormone receptor. Source data are provided as a Source Data file.

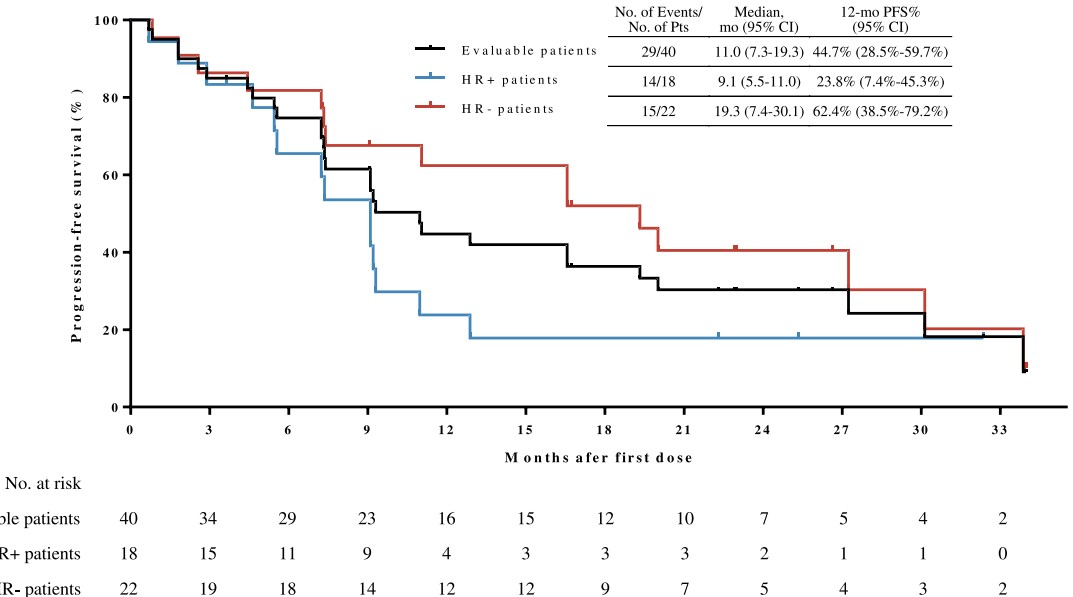

| | No. of Events/ No. of Pts | Median, mo (95% CI) | 12-mo PFS% (95% CI) |
|---|---|---|---|
| Evaluable patients | 29/40 | 11.0 (7.3-19.3) | 44.7% (28.5%-59.7%) |
| HR+ patients | 14/18 | 9.1 (5.5-11.0) | 23.8% (7.4%-45.3%) |
| HR- patients | 15/22 | 19.3 (7.4-30.1) | 62.4% (38.5%-79.2%) |

No. at risk

| | | | | | | | | | | | |
|---|---|---|---|---|---|---|---|---|---|---|---|
| Evaluable patients | 40 | 34 | 29 | 23 | 16 | 15 | 12 | 10 | 7 | 5 | 4 | 2 |
| HR+ patients | 18 | 15 | 11 | 9 | 4 | 3 | 3 | 3 | 2 | 1 | 1 | 0 |
| HR- patients | 22 | 19 | 18 | 14 | 12 | 12 | 9 | 7 | 5 | 4 | 3 | 2 |

**Fig. 3 | Kaplan-Meier estimates of progression-free survival in overall efficacy-evaluable population and subgroups of different HR status.** HR hormone receptor. Source data are provided as a Source Data file.

## Table 2 | Treatment-related adverse events

| | Any grade | Grade 3 | Grade 4 |
|---|---|---|---|
| Any event | 41 (100) | 34 (82.9) | 5 (12.2) |
| Diarrhea | 40 (97.6) | 9 (22.0) | 0 |
| White blood cells decreased | 40 (97.6) | 28 (68.3) | 0 |
| Neutrophil count decreased | 39 (95.1) | 22 (53.7) | 5 (12.2) |
| Anemia | 31 (75.6) | 8 (19.5) | 0 |
| Nausea | 22 (53.7) | 0 | 0 |
| Stomatitis | 21 (51.2) | 1 (2.4) | 0 |
| Lymphocyte count decreased | 16 (39.0) | 1 (2.4) | 0 |
| Creatinine increased | 16 (39.0) | 0 | 0 |
| Hypokalemia | 15 (36.6) | 0 | 0 |
| Hypertriglyceridemia | 14 (34.1) | 0 | 0 |
| Vomiting | 13 (31.7) | 0 | 0 |
| Blood lactate dehydrogenase increased | 13 (31.7) | 0 | 0 |
| Platelet count decreased | 11 (26.8) | 1 (2.4) | 0 |
| Alkaline phosphatase increased | 10 (24.4) | 1 (2.4) | 0 |
| Hyperglycemia | 10 (24.4) | 0 | 0 |
| Stomach pain | 8 (19.5) | 0 | 0 |
| Rash | 8 (19.5) | 0 | 0 |
| Aspartate aminotransferase increased | 8 (19.5) | 0 | 0 |
| α-hydroxybutyrate dehydrogenase increased | 7 (17.1) | 0 | 0 |
| Hypoalbuminemia | 7 (17.1) | 0 | 0 |
| Cholesterol high | 7 (17.1) | 0 | 0 |
| Pruritis | 7 (17.1) | 0 | 0 |
| Alanine aminotransferase increased | 6 (14.6) | 0 | 0 |
| Blood bilirubin increased | 6 (14.6) | 0 | 0 |
| Weight loss | 2 (4.9) | 1 (2.4) | 0 |
| Dermatitis | 2 (4.9) | 1 (2.4) | 0 |
| γ-glutamyltransferase increased | 2 (4.9) | 1 (2.4) | 0 |
| Ventricular arrhythmia | 1 (2.4) | 1 (2.4) | 0 |

Data are n (%).

## Safety

All the 41 patients who received at least one dose of the study drug had experienced at least one treatment-related adverse event (TRAE) during the study treatment, and 34 (82.9%) and five (12.2%) of them had TRAEs of grade 3 and 4, respectively. No grade 5 AEs were reported (Table 2). The most common TRAEs were diarrhea (40/41, 97.6%), decreased white blood cell (40/41, 97.6%), decreased neutrophil count (39/41, 95.1%), anemia (31/41, 75.6%), nausea (22/41, 53.7%), and stomatitis (21/41, 51.2%). The most common grade 3 TRAEs were decreased white blood cells (28/41, 68.3%), decreased neutrophil count (22/41, 53.7%), diarrhea (9/41, 22.0%), and anemia (8/41, 19.5%). Decreased neutrophil count was the only reported grade 4 TRAE, occurring in 12.2% of patients.

Dose reductions were required in 12 patients. Ten patients needed to reduce the dose of dalpiciclib to 100 mg/d, mostly due to decreased white blood cells and decreased neutrophil count, and three of them required a second dose reduction to 75 mg/d. Dose reductions of pyrotinib to 320 mg/d were recorded in four patients, and reasons for dose adjustments included diarrhea, weight loss, palmar-plantar erythrodysesthesia syndrome, decreased white blood cells, and decreased neutrophil count. There were two treatment-related serious adverse events (one abdominal pain and one decreased cardiac ejection fraction), and one of them (decreased cardiac ejection fraction) led to discontinuation of pyrotinib consequently. No treatment-related deaths occurred.

Of nine patients who experienced diarrhea of grade 3, two experienced more than once. The median time to the first onset of grade 3 diarrhea was 58 days (range = 3–231), and the median cumulative duration of grade 3 diarrhea was 2 days (range = 1–6). A dose reduction of pyrotinib related to diarrhea was reported in one patient. No dose discontinuation was related to diarrhea.

## Exploratory outcomes

Though not pre-planned, exploratory analyses were conducted in a post-hoc manner. By the data cutoff date, 30 patients (75.0%, 95% CI = 58.8–87.3%) achieved clinical benefits, and the median duration of response was 15.8 months (95% CI = 7.2–28.16) among the 28 responders.

In terms of subgroups, post-hoc exploratory analyses showed that ORR tended to be higher in patients with HR-negative disease (81.8% [95% CI = 59.7–94.8%], vs 55.6% [95% CI = 30.8–78.5%] in patients with HR-positive disease), and in trastuzumab-sensitive patients (80.0% [95% CI = 59.3–93.2%], vs 53.3% [95% CI = 26.6–78.7%] in trastuzumab-resistant patients; Supplementary Fig. 1). Patients with HR-negative or trastuzumab-sensitive disease tended to have longer median PFS (19.3 months [95% CI = 7.4–30.1] for HR-negative patients vs 9.1 months [95% CI = 5.5–11.0] for HR-positive patients; 12.9 months [95% CI = 7.2–NA] for trastuzumab-sensitive patients vs 9.1 months [95% CI = 1.8–16.6] for trastuzumab-resistant patients; Fig. 3 and Supplementary Fig. 2).

There were 13 patients with asymptomatic brain metastasis (BM) at baseline, and 12 of them had untreated BM. The overall ORR in these 13 patients was 84.6% (95% CI = 54.6–98.1%), and the central nervous system (CNS) response rate by Response Evaluation Criteria In Solid Tumors (RECIST) version 1.1 in patients with measurable intracranial lesions was 66.7% (6/9). The median PFS among all BM patients was 11.0 months (95% CI = 5.5–33.9; Supplementary Fig. 3). Five patients with isolated progression of brain lesions continued the study treatment after local stereotactic radiosurgery till the second progression, and the median time from the first dose to the second disease progression or death (PFS2) in the BM subgroup was 19.3 months (95% CI = 7.2–33.9). Given that the subgroup analyses were not pre-specified with a small sample size and insufficient power, the findings should be interpreted with caution.

## Discussion

DAP-HER-01 is a prospective study to report the activity and safety of a CDK4/6 inhibitor plus a HER2 TKI in patients with HER2-positive ABC regardless of HR status, and obtained positive results. Our results suggest that this chemo-free regimen has anti-tumor activity in HER2-positive ABC, including HR-negative, HER2-positive patients, and exhibits efficacy in patients with BM, providing an alternative fully oral regimen for patients with HER2-positive ABC.

Prior to our study, Goel et al.[13] and Ciruelos et al.[14] assessed the use of continuous low-dose ribociclib or a three-week regimen of 200 mg palbociclib combined with trastuzumab in HER2-positive BC including patients with HR-negative disease, respectively, but both studies showed limited activity, especially in HR-negative patients. It might be due to the fact that participants in these two studies were treated with trastuzumab and heavily pretreated. In the current phase 2 study, we enrolled HER2-positive patients with no more than one line of prior systemic treatment for advanced disease, and used dalpiciclib and pyrotinib as the therapeutic regimen. Our results showed an ORR of 70.0% and a median PFS of 11.0 months, which were consistent with the results in the pyrotinib plus capecitabine group in PHOEBE[9] and PHENIX[10] studies.

Pyrotinib has shown preliminary efficacy in trastuzumab-sensitive patients[9,10]. DAP-HER-01 enrolled 25 trastuzumab-sensitive patients, among whom the ORR of dalpiciclib and pyrotinib (80.0%) was comparable with that of docetaxel plus trastuzumab plus pertuzumab (THP) in CLEOPATRA (80.2%) and PUFFIN (79%), while the median PFS of 12.9 months was relatively shorter (vs. 14.5–18.7 months with THP)[15,16]. For the 15 patients with trastuzumab resistance in our study, the ORR was 53.3% and the median PFS was 9.1 months, similar to the results of trastuzumab emtansine (ORR = 35–43.6%, PFS = 6.8–9.6 months)[17,18], but inferior to trastuzumab deruxtecan (T-DXd; ORR = 79%, PFS = 28.2 months)[18]. We noted that there was a discrepancy between ORR and PFS. It might be partially because 92.7% of patients enrolled in DAP-HER-01 had visceral metastases, compared with a proportion of 67%–82% reported in aforementioned phase 3 trials[9,10,17–19], which might affect PFS outcome. In addition, exploring the mechanism underlying resistance to dalpiciclib and pyrotinib may help to better explain the outcome results. Activation of MAPK

signaling plays an essential role in HER2 resistance[5,20], and a translational work has recently reported that HER2-positive BC cells with MAPK pathway alternations are resistant to CDK4/6 inhibitors[20]. Thus, analyzing the potential mechanisms underlying tumor growth and progression during the treatment is needed in future studies.

It has been reported that HR status could affect the outcome of HER2-positive BC, and HR-negative subset usually benefits more from standard HER2-targeted therapy compared with HR-positive ones[9,10,16,18,19]. Consistently, the current study shows that HR-negative patients tend to achieve higher ORR (81.8% vs 55.6%) and longer PFS (19.3 vs 9.1 months), which is comparable with data of THP as first-line treatment in PUFFIN (ORR = 75.6%, PFS = 14.5 months)[15]. However, the inferior activity observed in the HR-positive subtype in our study might be attributed to the absence of inhibition on estrogen receptor signaling, which could be an escape mechanism to circumvent HER2 inhibition[21], thus might further contribute to the suboptimal PFS outcome in the overall population. Based on these data, we hypothesized that adding an anti-HER2 monoclonal antibody to dalpiciclib and pyrotinib might further prolong PFS in HR-negative and HER2-positive subset, while in HR-positive and HER2-positive cancer, the addition of endocrine therapy might be beneficial. Hence, we have designed another trial of first-line treatment for HER2-positive ABC (DAP-HER-02, NCT05328440), in which we use dalpiciclib and pyrotinib combined with fulvestrant (for HR-positive patients) or inetetamab (an anti-HER2 monoclonal antibody; for HR-negative patients), and the enrollment is ongoing.

BM occurred in 37.3% of patients with HER2-positive ABC, which suggests worse outcomes[22]. Most phase 3 trials excluded patients with BM (CLEOPATRA[16], PUFFIN[15], PHOEBE[9]) or only enrolled patients with treated and stable BM (EMILIA[17], DESTINY-Breast03[18]). Recently, it has been reported that the loss of p16[INK4A], the major endogenous CDK4/6 protein inhibitor, was found in a majority of HER2-positive BC BM patient-derived xenografts. The combination of CDK4/6 inhibitor abemaciclib and HER2 inhibitor tucatinib resulted in significant tumor regression in p16[INK4A]-negative, HER2-positive BC BM patient-derived xenograft models compared with either drug alone[23], suggesting that patients with HER2-positive BC and BM may benefit from the combination treatment. Our study included 13 patients (31.7%) with clinically asymptomatic BM and 12 of them never received radiotherapy or surgery for BM. Consistent with this preclinical observation, the median PFS in our BM population was 11.0 months, and CNS-ORR of patients with measurable intracranial diseases was 66.7% (6/9), which was similar to pyrotinib plus capecitabine in HER2-positive metastatic BC with radiotherapy-naïve BM (median PFS = 11.3 months, CNS-ORR = 74.6%)[24], and superior to other anti-HER2 regimens in active BMs (intracranial response rate = 11–57.1%[25–28], median time to progression = 5.5 months[27]). T-DXd is an emerging and promising HER2-target therapy in HER2-positive BC, and some small-sample trials have explored its efficacy in active BMs leading to inconsistent results. The primary analysis of the ongoing DEBBRAH trial[29] showed that the CNS-ORR of T-DXd in 13 BC patients with HER2-positive and active BM was 46.2% (50.0% in asymptomatic untreated BMs and 44.4% in progressing BMs after local therapy), and the PFS was immature. In TUXEDO-1 study[30], the CNS-ORR was 78.6% and the median PFS was 14 months in 14 per-protocol patients with active BMs. Though with limited cases, our findings indicate that the combination of dalpiciclib and pyrotinib is potentially an optional regimen for HER2-positive metastatic BC. It is worth further study in this field.

The safety profile of dalpiciclib and pyrotinib combination was consistent with the previous toxicity profile of either drug in previous clinical trials[8,9]. Diarrhea was the most common adverse event (AE) of pyrotinib, while the incidence and the median cumulative duration of grade ≥3 diarrhea was significantly reduced in this study (22.0% and 2 days), compared with pyrotinib combined with capecitabine (31% and 7.0 days in PHOEBE[9], 31.4% and 9.0 days in PHENIX[10]). Hematologic

toxicities of any grade and grade ≥3 in this study were similar to that of dalpiciclib plus fulvestrant in DAWNA-1[8], while severe decreased neutrophil count was less frequent, and all the hematologic toxicities were reversible with dose adjustments of dalpiciclib and symptomatic treatments. The incidence of AEs such as alopecia, fever, and peripheral edema was lower than that with THP or docetaxel plus trastuzumab reported in PUFFIN study[15]. In addition, the incidence of palmarplantar erythrodysesthesia syndrome, abnormal liver function, fatigue, and anorexia in this study was significantly lower than that with pyrotinib combined with chemotherapy[9,10]. The results suggest that the safety profile of this dual-combination is manageable, and our regimen may be an alternative option for patients who would mind or previously show chemotherapy toxicities such as alopecia, fatigue, and hepatic dysfunction.

There are some limitations in this study. Firstly, the current study adopted single-arm design with limited sample size, and only Chinese patients were enrolled. Secondly, biomarker analysis might provide more information to identify who can derive more benefits from this combination, but such analysis was absent due to insufficient amount of collected tumor tissues and limited financial resources. Thirdly, though dose schedules used in the study were based on the approved doses and safety profiles derived from studies of these two agents, a dose exploration of this combination might be more convincing. Fourthly, though the activity of dalpiciclib and pyrotinib in patients with HR-negative disease and in those with BM was promising, the findings should be interpreted with cautions since the subgroup analyses were not pre-specified or powered, and further investigation is needed.

Overall, the combination of dalpiciclib plus pyrotinib shows promising activity and manageable toxicity in the front-line treatment of HER2-positive ABC patients regardless of HR status, and patients with asymptomatic active BM may also benefit from it. The inaccessibility of study drugs in some countries and regions, coupled with the plethora of anti-HER2 therapies available, may constrain the generalizability of our findings in clinical settings. Nevertheless, this fully oral regimen provides a convenient alternative option for patients with HER2-positive ABC. Furthermore, this study may represent a direction for further exploration into the role of CDK4/6 inhibitors in BC patients.

## Methods

The study was conducted in accordance with the International Conference on Harmonization Good Clinical Practice Guidelines (ICH GCP) and the Declaration of Helsinki, and was approved by Ethics Committees from Henan Cancer Hospital. All patients provided written informed consent before the initiation of any study-related treatment or procedures. This trial was registered at www.clinicaltrials.gov (ClinicalTrials.gov identifier: NCT04293276) on March 3, 2020.

### Patients

DAP-HER-01 (NCT04293276) was a single-arm, phase 2 clinical trial conducted in the Affiliated Cancer Hospital of Zhengzhou University/ Henan Cancer Hospital, Zhengzhou, China.

The enrollment was conducted between April 9, 2020 and May 19, 2021. Women aged 18–70 years with histologically confirmed advanced HER2-positive breast cancer defined as IHC score of 3 + , or IHC score of 2+ and positive FISH test according to the 2018 American Society of Clinical Oncology (ASCO)/College of American Pathologists (CAP) guidelines, previously treated with no more than one systemic therapy in advanced setting were eligible for this trial. Further eligibility criteria included at least one measurable lesion based on RECIST 1.1, an Eastern Cooperative Oncology Group (ECOG) performance status of 0 or 1, a life expectancy of at least 12 weeks, and adequate organ function. Prior trastuzumab was allowed. Women with childbearing potential must have a negative pregnancy test at screening.

Patients were excluded if they had symptomatic BM, or previously received treatment with any TKI targeting HER2 or CDK4/6 inhibitor. Anti-cancer treatment (radiotherapy, chemotherapy, major surgery, or targeted therapy) was not permitted within four weeks before enrollment, except for endocrine therapy, which should be discontinued during the screening period. Patients with concomitant diseases that seriously endanger the safety or affect the completion of the study were excluded, and judged by investigators.

### Study design and treatment

All patients received dalpiciclib 125 mg daily orally for 21 days followed by seven days off and pyrotinib 400 mg daily orally in each 28-day cycle. Treatment was continued until disease progression, unacceptable toxicity, death, consent withdrawal, or investigator decision. Concomitant endocrine therapy was not allowed. Participants with BM at baseline developing isolated intracranial progression could continue study treatment after brain radiotherapy till the second progression if they would benefit from the treatment according to the physicians' discretion. Dose interruptions and reductions were allowed to manage AEs. Dose adjustments for dalpiciclib and pyrotinib were permitted as specified in the protocol. The trial protocol is available in the Supplementary Information.

### Endpoints and assessment

The primary endpoint was ORR (the proportion of patients with the best overall response of complete or partial response). Secondary endpoints included PFS (defined as the time from the first dose of study treatment to the documented disease progression or death due to any cause, whichever occurred first; the second progression or death in patients with isolated intracranial progression who continued the study treatment after radiotherapy would be counted as PFS2 event that analyzed in patients with BM), OS (defined as the time from the first dose of study treatment to death due to any cause), and safety.

Tumor assessments were done by contrast-enhanced computed tomography or magnetic resonance imaging (MRI; brain metastases were assessed by MRI) at baseline, every two cycles for the first 18 cycles, and every three cycles thereafter. Complete or partial response would be confirmed 4–6 weeks later. Bone scans were conducted when lesions or disease progression in bone were suspected. ORR and PFS were assessed by investigators based on RECIST 1.1. Safety was assessed with laboratory assessments, 12-lead electrocardiograms, echocardiograms, physical examinations, and AEs graded according to the National Cancer Institute Common Terminology Criteria for Adverse Events (NCI CTCAE; version 5.0).

### Statistical analysis

A Simon minimax two-stage design was used to estimate the sample size, with a one-sided alpha of 5% and power of 80%. The null hypothesis (P0) was set as an ORR of 50%, referring to the ORR of pyrotinib monotherapy in HER2-positive ABC patients in a phase I study[31]. Since the ORR was 68.6% with the treatment of pyrotinib and capecitabine in PHINEX[10] study, the expected ORR (P1) was set as 70%. Accordingly, the first stage was planned to enroll 23 patients. If more than 12 of these patients achieved response, 14 additional patients would be recruited in the second stage. The study would meet its primary endpoint if confirmed objective response was observed in 24 or more patients out of a total of 37 response-evaluable patients. Considering a dropout rate of 10%, 41 patients would be enrolled.

Data were collected using Electronic Data Capture System. Efficacy analyses were performed in eligible participants who received at least one efficacy evaluation, and safety analyses were carried out in all patients who received at least one dose of study treatment. The primary endpoint of ORR was estimated with a 95% confidence interval (CI). PFS and OS were estimated using the Kaplan-Meier method. All statistical analyses were performed using SAS (version 9.2).

**Reporting summary**

Further information on research design is available in the Nature Portfolio Reporting Summary linked to this article.

## Data availability

The raw clinical data are protected and are not available due to data privacy laws. The de-identified datasets supporting the findings of this study are available for academic purposes on request from the corresponding author, Min Yan (ym200678@126.com) for 5 years, with the approval of the Institutional Ethical Committees. The trial protocol is available as Supplementary Note in the Supplementary Information. Source data are provided in this paper. The remaining data are available within the Article and Supplementary Information.

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

## Acknowledgments

We would like to acknowledge the patients and their families, the study investigators, and the clinical site staff. As an investigator-initiated trial, the authors acknowledge Henan Cancer Hospital for providing research facilities and equipment. The study drugs, dalpiciclib and pyrotinib,

were provided by Jiangsu Hengrui Pharmaceuticals Co., Ltd. This study was funded by the XINRUI Project of Cancer Supportive Care and Treatment Research (cphcf-2021-011, M.Y.).

## Author contributions

We had full access to all data in the trial and took responsibility for the integrity of the data and the accuracy of the data analysis. M.Y. contributed to the conceptualization and design of the trial. L.N., H.L., M.Z., Z.Z.L., X.C., Z.D.L., C.Z., H.Z., S.Z., Y.F., H.S., J.W. and H.L. were responsible for the collection and assembly of data. J.W. and L.N. completed the statistical analyses. All authors participated in writing the paper and approved the final version of the paper.

## Competing interests

H.L. was an employee of Jiangsu Hengrui Pharmaceuticals Co., Ltd (Shanghai, China) during the study period. The remaining authors have no conflicts of interest to declare.
