## [Peer Review File · Nature Communications]

Reviewers' Comments:

Reviewer #1:

Remarks to the Author:

Introduction of the review and general comments:

This is a well-conducted phase 2 trial exploring the activity and safety of dalpiciclib in combination with pyrotinib in HER2-positive (+) metastatic breast cancer who had received maximum one line of therapy in metastatic setting. This study has been conducted without prior published data on the safety of the combination and neither a dose escalation of the two drugs for the combination. Authors reported this point in the discussion as weakness of the trial. In my opinion, in this setting it would have been appropriate including safety as primary endpoint. Indeed, due to the small sample size and the trial design (single arm, non-randomized), this trial could not be a pivotal and conclusive trial in terms of efficacy but represents a solid rationale for larger studies (already ongoing). Nevertheless, the two drugs have been largely investigated in the context of other regimens, and the absence of overlap toxicities, made this choice acceptable from safety standpoint.

Key results:

This study is the first clinical trial evaluating a new generation CDK4/6 inhibitor in combination a pan-HER2 tyrosine kinase inhibitor in metastatic HER2-positive breast cancer. The key results include ORR of 70% and 11 months of progression free-survival (PSF), numerically driven by hormone receptor negative (HR-)/HER2+ population. This result is relevant if we think that is a chemo-free and oral regimen. Interestingly, patients with brain metastases at baseline (N=13, 12 of them with active and asymptomatic) had a CNS-ORR of 66.7% (6/9 evaluable patients), confirming the activity of pyrotinib in the CNS compartment, even without chemotherapy backbone. In terms of safety, 100% of patient experienced at least 1 side effect with bone marrow toxicity and diarrhea as prevalent (97.6 % in both cases), related to dalpiciclib and pyrotinib, respectively.

Validity:

As previously stated, this trial has a daring design using efficacy as primary endpoint and provides a signal of activity of the combination. The sample size is too small to be able to demonstrate the benefit according to HR status and trastuzumab resistance. The benefit (ORR and PFS) is higher in the HR- group, but it is difficult to derive if this difference has also predictive significance rather than just prognostic, because the study is underpowered to reply to this question. I found very interesting the choice of using CDK4/6 inhibitor instead of chemotherapy and in absence of endocrine backbone as a legitimization of the use of this class of drug in HER2-pathway addicted tumors regardless the estrogen signaling. This point is mentioned in the introduction (references: Sinclair et al. Clin Breast Cancer, 2022; Goel et al, Cancer Cell 2016) but not in the discussion. If this superior activity observed in HR- over HR+ is related to the upregulation of pRb or other biomarkers is unknown. Biomarker analysis (liquid biopsy and/or genomics) might help to clarify which group of patients can derive more benefit from this new combination. Being aware of status of PI3K pathway alterations as well as MAPK pathway, both known mechanism of resistance of trastuzumab and pertuzumab can be useful to better interpreted the outcome results, but also can allow to properly collocate/sequence this regimen in the anti-HER2 therapy portfolio. A translational work conducted at Memorial Sloan Kettering Cancer Center showed that HER2+ breast cancer cells with MAPK pathway alternations (including NF1 loss, HER2 activating mutations and others) are resistant to CDK4/6 inhibitors (Palbociclib) but are sensitive to CDK2 or CDK2/4/6 inhibition. It would be interesting to look at the MAPK alterations in this cohort of patients treated with a new potent CDK4/6 I and panHER2 TKI Smith A & Chandarlapaty S, Nat Comm, 2021). With respect to the trastuzumab resistant/sensitivity, I am not sure this definition can apply in this context. Indeed, HER2+ tumors are highly heterogenous and it is known that at least part of the cells, preserve trastuzumab sensitivity even after early progression to trastuzumab. Therefore, I would consider rephrasing as "early progression to trastuzumab" or similar. The reported toxicity of the combination was acceptable and dose reduction was required in 12/41 (30%). To be noted that, only patients not eligible for chemotherapy or patients who declined chemotherapy are included in this trial. This inclusion criteria introduced a selection bias and the toxicity reported may be overestimated, especially if most of the patients enrolled were not eligible for chemo. I suggest to the authors, if available, to add in the supplementary section and/or

comment in the discussion, the proportion of patients who declined chemo versus not eligible for chemo, and in the latter group explain why they were not eligible.

Significance:

This study is a proof-of-concept trial that can be the rationale of future trials design, perhaps also in early stage. This trial provides supporting data in terms of efficacy and safety in first/second line regimen, but in my opinion not conclusive to be translate directly in clinical practice. With the extreme active regimens in second line like T-DXd and tucatinib (validated on phase II-III larger randomized trials). I would be cautious to use this regimen in clinical practice. Indeed, both T-DXd and HER2Climb regimens are well tolerated, and they are not associate to the classical chemotherapy side effects. In addition, median PFS of 11 months is shorter than expected in a combination with 70% of ORR and it is shorter than other options in the same setting. In addition, despite both pyrotinib and dalpiciclib showed extremely promising results, both are tested selectively in a Chinese population (non-US FDA approved, so far) and this represents a significant barrier at this time to further develop the combination with worldwide accessibility.

Data and methodology:

From my prospective (clinician), the data are well analyzed and presented. Given the study did not have a hypothesis based on subgroups HR-positivity/negative and trastuzumab sensitive/resistant, I would suggest specifying that difference seen in these subgroups, although relevant from clinical standpoint, has been reported as descriptive and results should be interpreted with caution (study underpowered). For transparency, I suggest reporting (at least as supplementary material) the interim analysis that should have been done to achieve the full accrual according to the Simon two stage design.

Analytical approach:

As the authors reported in the discussion, the main limitations include the small sample size, single arm design, restriction to Chinese population. The fact that only 12% of patients received prior pertuzumab, make this population different from patients we use to follow in western countries that are all pretreated with double blockade in early and/or metastatic setting. There is absence of safety of the combination and no preclinical model that suggest a synergic activity of the two agents as well as no dose escalation trial. We cannot exclude that a different schedule/dose can be more effective in this setting (I would mention this in the discussion).

Suggested improvements:

See other sections above. I suggest emphasizing in the discussion that this is the first study with CDK4/6i and HER2 TKI without chemo backbone with promising results. They could mention the phase study with ribociclib and trastuzumab that provided disappointing results (Goel et al. Clinical Breast Cancer 2019

<https://www.sciencedirect.com/science/article/pii/S1526820919303131?via=ihub#bib10>).

In addition, in the method section, I would clearly that concomitant endocrine therapy was not allowed since this element is important for the results interpretation.

Clarity and context:

The language of the work is appropriated, and the results are presented clearly. References are properly reported.

Reviewer #2:

Remarks to the Author:

The authors report on a phase II study of dalpiciclib and pyrotinib, CDK4/6i inhibitors, in 41 HER2+ advanced breast cancer patients. Primary efficacy endpoint was met with an estimated 70% objective response. The dual regimen was safe with manageable AEs and no deaths.

Comments:

1. In the introduction, the novelty is couched as the use of the dual regimen in HER2+, HR- patients. However, the protocol population was in HER2+ patients regardless of HR status. Is this the main novelty or is there more novelty that could be highlighted?

2. My calculations show 24 or more responses (not 23 or more) for a successful Simon 2-stage minimax design.
3. The protocol full analysis set describes all enrolled patients who received at least one dose of study drug to be used for efficacy. However, the study reports efficacy out of 40 patients. Figure 1 legend notes that one patient received study treatment but was excluded due to not meeting inclusion criteria. If they did not meet inclusion criteria, why were they treated? More details should be provided about this discrepancy. The audience will want to know. It is bizarre.
4. The authors report subgroup analyses which were not pre-specified or powered. This should be stated. Ideally the authors would formally statistically test each interaction in one model rather than performing subgroup analyses which are generally ill-advised.
<https://pubmed.ncbi.nlm.nih.gov/15066682/>
<https://pubmed.ncbi.nlm.nih.gov/25030633/>
<https://pubmed.ncbi.nlm.nih.gov/10744093/>
5. Figure 2b only shows 39 patients. An HR+ patient is missing. Why?
6. Figure 3: x axis could show 0-100% to be more aligned with actual estimates in the figure.
7. Figure 3: symbol typo for HR less than or equal to 10%.
8. Figure 4, Extended Figures 1 and 2—please provide confidence intervals for 12-mo PFS estimates.
9. 12 and 18-month OS estimates are provided in the text. Please give confidence intervals for these estimates.
10. The authors discuss the patients with brain mets in their study. The authors should note that these patients did not have clinically symptomatic brain mets which was an exclusion criterium in their protocol. The conclusion that dalcipicilicilb and pyrotinib may be an active regimen for HER2+ mBC with active BMs needs to be modified for transparency.

Reviewer #3:

Remarks to the Author:

The authors of this study are to be congratulated for designing and conducting a study that is supported by a sound preclinical rationale. However, the study shows weaknesses that preclude its completeness and generalizability.

First, it is a single-arm study, with 32% of enrolled patients being anti-HER2 naive.

Second, there is no attempt to characterize responders although the study population is very small (n=40). It would be important to have information on any biomarkers that could identify which patients are most likely to benefit from a combination of CDK4/CDK6 and HER2 inhibition. Not least because, authors offer a counterintuitive finding to previous studies, as patients with HR-negative tumors appear to be the most responsive to treatment.

The data on patients with brain metastasis (31% of the whole study population) is interesting given the tendency of HER2-positive breast cancer to involve the brain. However, the study was not designed or powered to investigate this issue, neither the authors comment why median PFS was overall less than one year, which is quite disappointing when we consider the current standard of care and those soon to come.

The discussion is verbose and authors report comparisons with existing literature without making any effort to point out the difference in study design, and population size and characteristics among the cited clinical trials.

REVIEWER COMMENTS

Reviewer #1 - Breast cancer clinical trials

Introduction of the review and general comments:

This is a well-conducted phase 2 trial exploring the activity and safety of dalpiciclib in combination with pyrotinib in HER2-positive (+) metastatic breast cancer who had received maximum one line of therapy in metastatic setting. This study has been conducted without prior published data on the safety of the combination and neither a dose escalation of the two drugs for the combination. Authors reported this point in the discussion as weakness of the trial. In my opinion, in this setting it would have been appropriate including safety as primary endpoint. Indeed, due to the small sample size and the trial design (single arm, non-randomized), this trial could not be a pivotal and conclusive trial in terms of efficacy but represents a solid rationale for larger studies (already ongoing). Nevertheless, the two drugs have been largely investigated in the context of other regimens, and the absence of overlap toxicities, made this choice acceptable from safety standpoint.

Key results:

This study is the first clinical trial evaluating a new generation CDK4/6 inhibitor in combination a pan-HER2 tyrosine kinase inhibitor in metastatic HER2-positive breast cancer. The key results include ORR of 70% and 11 months of progression free-survival (PSF), numerically driven by hormone receptor negative (HR-)/HER2+ population. This result is relevant if we think that is a chemo-free and oral regimen. Interestingly, patients with brain metastases at baseline (N=13, 12 of them with active and asymptomatic) had a CNS-ORR of 66.7% (6/9 evaluable patients), confirming the activity of pyrotinib in the CNS compartment, even without chemotherapy backbone. In terms of safety, 100% of patient experienced at least 1 side effect with bone marrow toxicity and diarrhea as prevalent (97.6 % in both cases), related to dalpiciclib and pyrotinib, respectively.

Response: We greatly appreciate your thoughtful review and insightful comments on our study. As you have mentioned, since the safety of these two drugs have been largely investigated in other

studies which showed no overlapping toxicities, we only designed safety as a secondary endpoint in this study. Although we cannot change it to a primary endpoint in this study, we have included a detailed analysis of safety.

Validity:

As previously stated, this trial has a daring design using efficacy as primary endpoint and provides a signal of activity of the combination. The sample size is too small to be able to demonstrate the benefit according to HR status and trastuzumab resistance. The benefit (ORR and PFS) is higher in the HR- group, but it is difficult to derive if this difference has also predictive significance rather than just prognostic, because the study is underpowered to reply to this question.

Response:

Thanks for your comments. The initial study aims to investigate whether the benefit of the combination of dalpiciclib and pyrotinib can expand to all HER2-positive BCs. Thus, the subgroup analyses were not pre-specified or powered. We cannot determine HR status as a clinical benefit predictor, owing to small size sample, as you pointed out. Nevertheless, we wish to truthfully report such interesting observation in the current study, as it may inspire future studies. We are also conducting a follow-up study, DAP-HER-02, to make up for this regret.

To avoid any misunderstandings, we have added this limitation to the text (Line 206-209):

Fourthly, though the efficacy of dalpiciclib and pyrotinib in patients with HR-negative disease and in the ones with brain metastases was promising, the findings should be interpreted with caution since the subgroup analyses were not pre-specified or powered, and further investigation is needed.

I found very interesting the choice of using CDK4/6 inhibitor instead of chemotherapy and in absence of endocrine backbone as a legitimization of the use of this class of drug in HER2-pathway addicted tumors regardless the estrogen signaling. This point is mentioned in the introduction (references: Sinclair et al. Clin Breast Cancer, 2022; Goel et al, Cancer Cell 2016) but not in the discussion. If

this superior activity observed in HR- over HR+ is related to the upregulation of pRb or other biomarkers is unknown. Biomarker analysis (liquid biopsy and/or genomics) might help to clarify which group of patients can derive more benefit from this new combination. Being aware of status of PI3K pathway alterations as well as MAPK pathway, both known known mechanism of resistance of trastuzumab and pertuzumab can be useful to better interpreted the outcome results, but also can allow to properly collocate/sequence this regimen in the anti-HER2 therapy portfolio. In a translational work conducted at Memorial Sloan Kettering cancer center, we reported that HER2+ breast cancer cells with MAPK pathway alternations (including NF1 loss, HER2 activating mutations and others) are resistant to CDK4/6 inhibitors (Palbociclib) but are sensitive to CDK2 or CDK2/4/6 inhibition. It would be interesting to look at the MAPK alterations in this cohort of patients treated with new potent CDK4/6 I and panHER2 TKI (Smith A & Chandarlapaty S, Nat Comm, 2021).

Response: Thanks for your constructive advice. Unfortunately, we are unable to conduct the biomarker analysis owing to insufficient amount of tissue sample collected through puncture biopsy of advanced stage patients after various diagnostic tests needed for treatment, and limited financial resources in this study.

We have discussed this point in the limitations of this study in the revision (Line 201-204):

Secondly, biomarker analysis might provide more information to identify who can derive more benefit from this combination, but such analysis was absent due to insufficient amount of collected tumor tissue and limited financial resources.

Furthermore, we agree with you that analyzing the MAPK pathway alteration to explore the mechanism underlying resistance to darpiciclib and pyrotinib in future studies may help to better explain the outcome results. We have added this point in the Discussion and the reference you mentioned has been cited. [Line 140-146, Reference No. 20: Smith AE, et al. HER2 + breast cancers evade anti-HER2 therapy via a switch in driver pathway. Nat Commun 12, 6667 (2021)]:

In addition, exploring the mechanism underlying resistance to darpiciclib and pyrotinib may help

to better explain the outcome results. Activation of MAPK signaling plays an essential role in HER2-resistance^{5,20}, and a translational work has recently reported that HER2-positive breast cancer cells with MAPK pathway alternations are resistant to CDK4/6 inhibitors²⁰. Thus, analyzing the alteration of MAPK pathway and other signaling related to tumor growth and progression during the treatment is needed in future studies.

With respect to the trastuzumab resistant/sensitivity, I am not sure this definition can apply in this context. Indeed, HER2+ tumors are highly heterogenous and it is known that at least part of the cells, preserve trastuzumab sensitivity even after early progression to trastuzumab. Therefore, I would consider rephrasing as “early progression to trastuzumab” or similar.

Response: Thanks for your suggestion. We totally agree with you that “trastuzumab resistant” does not mean all cells lost trastuzumab sensitivity. We chose this terminology to be consistent with the convention in other clinical studies such as the PHOEBE study¹, the PANACEA study², and the PERMEATE study³, where the term “trastuzumab resistant” was indeed used in a similar context. We agree, however, that it is a somewhat inaccurate term to describe such situation, thus a more accurate term could be chosen as the convention in the future, as you have recommended.

References:

1. Xu B, et al. Pyrotinib plus capecitabine versus lapatinib plus capecitabine for the treatment of HER2-positive metastatic breast cancer (PHOEBE): a multicentre, open-label, randomised, controlled, phase 3 trial. *Lancet Oncol* 22, 351-360 (2021).
2. Loi S, et al. Pembrolizumab plus trastuzumab in trastuzumab-resistant, advanced, HER2-positive breast cancer (PANACEA): a single-arm, multicentre, phase 1b-2 trial. *Lancet Oncol* 20, 371-382 (2019).
3. Yan M, et al. Pyrotinib plus capecitabine for patients with human epidermal growth factor receptor 2-positive breast cancer and brain metastases (PERMEATE): a multicentre, single-arm, two-cohort, phase 2 trial. *Lancet Oncol* 23, 353-361 (2022).

The reported toxicity of the combination was acceptable and dose reduction was required in 12/41 (30%). To be noted that, only patients not eligible for chemotherapy or patients who declined chemotherapy are included in this trial. This inclusion criteria introduced a selection bias and the toxicity reported may be overestimated, especially if most of the patients enrolled were not eligible for chemo. I suggest to the authors, if available, to add in the supplementary section and/or comment in the discussion, the proportion of patients who declined chemo versus not eligible for chemo, and in the latter group explain why they were not eligible.

Response: Thanks for your kind advice and sorry for the confusion. Actually, all patients enrolled were eligible for chemotherapy, in terms of the patients' physical conditions according to the inclusion criteria. The main reason limiting the patients from undergoing chemotherapy was due to Covid-19 policy restrictions in China at the time of the study (April 2020 to May 2021). Therefore, we think there is no selection bias in this study. We do see that using "not suitable for or rejecting chemotherapy" to describe the situation mentioned above may cause confusion for readers. After careful evaluation and confirming that it would not affect the scientific validity of the study, we decided to delete such a description from the manuscript.

Significance:

This study is a proof-of-concept trial that can be the rationale of future trials design, perhaps also in early stage. This trial provides supporting data in terms of efficacy and safety in first/second line regimen, but in my opinion not conclusive to be translate directly in clinical practice. With the extreme active regimens in second line like T-DXd and tucatinib (validated on phase II-III larger randomized trials). I would be cautious to use this regimen in clinical practice. Indeed, both T-DXd and HER2Climb regimens are well tolerated, and they are not associate to the classical chemotherapy side effects. In addition, median PFS of 11 months is shorter than expected in a combination with 70% of ORR and it is shorter than other options in the same setting. In addition, despite both pyrotinib and dalpiciclib

showed extremely promising results, both are tested selectively in a Chinese population (non-US FDA approved, so far) and this represents a significant barrier at this time to further develop the combination with worldwide accessibility.

Response: Thanks for the comments. We agree that the applicability of our findings to clinical practice is limited. Nevertheless, we still see our study as valuable step, as this fully oral regimen provides a convenient treatment option for HER2-positive ABC, and our study brings a new direction for research on the CDK4/6 inhibitors in breast cancer patients. We have included a comprehensive summary of the clinical significance and research value of the study results in the last paragraph of the Discussion (Line 212-217):

The inaccessibility of study drugs in some countries and regions, coupled with the plethora of anti-HER2 therapies available, may constrain the generalizability of our findings in clinical settings. Nevertheless, this fully oral regimen provides a convenient alternative for patients with HER2-positive ABC. Furthermore, this study may represent a new direction for further exploration into the role of CDK4/6 inhibitors in breast cancer patients.

Data and methodology:

From my prospective (clinician), the data are well analyzed and presented. Given the study did not have a hypothesis based on subgroups HR-positivity/negative and trastuzumab sensitive/resistant, I would suggest specifying that difference seen in these subgroups, although relevant from clinical standpoint, has been reported as descriptive and results should be interpreted with caution (study underpowered).

Response: Thank you for the suggestion. We have added the statement accordingly in the efficacy paragraph (Line 70-71) and limitations paragraph (Line 207-209):

In terms of subgroups, post-hoc exploratory analyses showed that ORR tended to be higher in patients with HR-negative disease...

the findings should be interpreted with caution since the subgroup analyses were not pre-specified

or powered, and further investigation is needed.

For transparency, I suggest reporting (at least as supplementary material) the interim analysis that should have been done to achieve the full accrual according to the Simon two stage design.

Response: Thanks for your suggestion. We have incorporated the enrolled number and ORR results of the first phase of Simon design in the first paragraph of the Results (Line 56-57):

As per protocol, 24 patients were enrolled in the first stage and 17 responses were achieved, reaching the preset target for this stage, so the recruitment continued.

Analytical approach:

As the authors reported in the discussion, the main limitations include the small sample size, single arm design, restriction to Chinese population. The fact that only 12% of patients received prior pertuzumab, make this population different from patients we use to follow in western countries that are all pretreated with double blockade in early and/or metastatic setting.

Response: Thanks for the comments. Unfortunately, there is still a significant proportion of BC patients in China who are unable to receive HER2-targeted treatment in the first-line due to the poor accessibility of pertuzumab and financial burden.

There is absence of safety of the combination and no preclinical model that suggest a synergic activity of the two agents as well as no dose escalation trial. We cannot exclude that a different schedule/dose can be more effective in this setting (I would mention this in the discussion).

Response: Thanks for your comment.

Preclinical studies have confirmed that daltapiciclib and pyrotinib have synergistic effects in HR+ and HER2+ breast cancer models¹, as we have mentioned in the introduction (Line 46-48).

In addition, our dosage selection was based on previous studies on the two drugs^{2,3}, which have already undergone dose escalation, and concluded with an approved dosage of 150 mg for dalpiciclib and 400 mg for pyrotinib. Considering the high proportion of Grade 3 or above neutropenia events with the standard dose of dalpiciclib, we reduced it to 125 mg in our study. Even so, a dose escalation trial may provide more convincing evidence that the dose used in this study is appropriate for this setting. And we have described it in the limitations (Line 204-206):

Thirdly, though dose schedules used in the study were based on the approved doses and safety profile derived from studies of these two agents, a dose exploration of this combination might be more convincing.

References:

1. Wang Y, et al. The Synergistic Effects of SHR6390 Combined With Pyrotinib on HER2+/HR+ Breast Cancer. *Front Cell Dev Biol* 9, 785796 (2021).
2. Zhang P, et al. A phase I study of dalpiciclib, a cyclin-dependent kinase 4/6 inhibitor in Chinese patients with advanced breast cancer. *Biomark Res* 9, 24 (2021).
3. Li Q, et al. Safety, Efficacy, and Biomarker Analysis of Pyrotinib in Combination with Capecitabine in HER2-Positive Metastatic Breast Cancer Patients: A Phase I Clinical Trial. *Clin Cancer Res* 25, 5212-5220 (2019).

Suggested improvements:

See other sections above. I suggest emphasizing in the discussion that this is the first study with CDK4/6i and HER2 TKI without chemo backbone with promising results. They could mention the phase study with ribociclib and trastuzumab that provided disappointing results (Goel et al. *Clinical Breast Cancer* 2019 <https://www.sciencedirect.com/science/article/pii/S1526820919303131?via=ihub#bib10>).

Response: Thank you so much for your kind and helpful advice. We have incorporated this point to enrich the revised Discussion (Line 117-119, Line 122-126):

To our knowledge, DAP-HER-01 is the first prospective study to report the activity and safety of

CDK4/6 inhibitors and HER2 TKIs without a chemo-backbone in HER2-positive ABC regardless of HR status that also obtained positive results.

Prior to our study, Goel S, et al.¹³ and Ciruelos E, et al.¹⁴ have assessed the use of continuous low-dose ribociclib or a three-week regimen of 200 mg palbociclib combined with trastuzumab in HER2-positive breast cancer including patients with HR-negative diseases, but both studies showed limited activity, especially in HR-negative patients. It may be due to the fact that participants in these two studies were treated with trastuzumab and heavily pretreated.

In addition, in the method section, I would clearly that concomitant endocrine therapy was not allowed since this element is important for the results interpretation.

Response: Thanks for your advice. We have added the relevant description in the 'Study design and treatment' section of Methods (Line 246-247).

... Concomitant endocrine therapy was not allowed.

Clarity and context:

The language of the work is appropriated, and the results are presented clearly.

References are properly reported.

Response: Thanks for recognizing our work.

Reviewer #2 - Biostatistics, clinical trial design:

The authors report on a phase II study of dalpiciclib and pyrotinib, CDK4/6i inhibitors, in 41 HER2+ advanced breast cancer patients. Primary efficacy endpoint was met with an estimated 70% objective response. The dual regimen was safe with manageable AEs and no deaths.

Comments:

1. In the introduction, the novelty is couched as the use of the dual regimen in HER2+, HR- patients. However, the protocol population was in HER2+ patients regardless of HR status. Is this the main novelty or is there more novelty that could be highlighted?

Response: Thanks for your comments which make us realize that we did not clearly state the novelty of this study in the introduction. The novelty of this study was concluded as below.

- 1) Patients in this study were not limited to the HR-positive, but also the HR-negative with HER2-positive BC.
- 2) The HER2-targeted agent used in this study was a TKI, not a mono-antibody such as trastuzumab which has been previously explored with CDK4/6 inhibitor.
- 3) The combination strategy in our study was without chemo- or endocrine- backbone in HER2-positive breast cancer.

We have modified the description to show the novelty of this study in the Introduction (Line 37-40, Line 50-52):

*Clinical trials have demonstrated the efficacy of the combination of CDK4/6 inhibitors and HER2-targeted agents in HR-positive and HER2-positive advanced BC^{6,7}. However, **whether the benefit of the combination can expand to all HER2-positive BCs requires further exploration.***

...

*This is the first study exploring the combination of **a CDK4/6 inhibitor and a HER2 TKI without a chemo- or an endocrine- backbone** in HER2-positive breast cancer.*

2. My calculations show 24 or more responses (not 23 or more) for a successful Simon 2-stage minimax design.

Response: Thanks a lot for pointing out the mistake. We apologize for the clerical error. As per protocol (see Supplementary Information- Supplementary Note 1. Study protocol-page 5), if more than 23 subjects achieve CR/PR, the treatment plan will be considered to be effective, we have corrected it in our revised manuscript (Line 275):

...if confirmed responses were observed in 24 or more patients out of a total of 37 response-evaluable patients.

3. The protocol full analysis set describes all enrolled patients who received at least one dose of study drug to be used for efficacy. However, the study reports efficacy out of 40 patients. Figure 1 legend notes that one patient received study treatment but was excluded due to not meeting inclusion criteria. If they did not meet inclusion criteria, why were they treated? More details should be provided about this discrepancy. The audience will want to know. It is bizarre.

Response: Thanks for your suggestion. We apologize that we did not clarify the exclusion reason. Patients included in the study should be identified with an immunohistochemistry (IHC) score of 3+ or IHC 2+ and ISH/FISH+. However, we found the patient was with HER2 IHC 1+ after receiving study treatment, thus we excluded her from efficacy analysis.

We have explained the reason in Figure legend (Line 425-426) and in the first paragraph of Results (Line 58-62):

*Figure 1. Trial profile. * One patient was found with HER2 IHC 1+ after receiving study treatment and was excluded from efficacy analysis due to not meeting the inclusion criteria.*

One patient's updated information showed she had a HER2 immunohistochemistry (IHC) score of 1+ and HER2 gene amplification by fluorescence in-situ hybridization (FISH), which did not meet the inclusion criteria. Thus, she was excluded from the efficacy analysis though she received study treatment (Fig. 1).

4. The authors report subgroup analyses which were not pre-specified or powered. This should be stated. Ideally the authors would formally statistically test each interaction in one model rather than performing subgroup analyses which are generally ill-advised.

<https://pubmed.ncbi.nlm.nih.gov/15066682/>

<https://pubmed.ncbi.nlm.nih.gov/25030633/>

<https://pubmed.ncbi.nlm.nih.gov/10744093/>

Response: Thanks for your comments. We totally agree with you and have stated that the subgroup analyses were not pre-specified or powered, and should be interpreted with caution in the Results (Line 70-71) and Discussion (206-209), accordingly.

In terms of subgroups, post-hoc exploratory analyses showed that ORR tended to be higher...

Fourthly, though the efficacy of daltapiciclib and pyrotinib in patients with HR-negative disease and in the ones with brain metastases was promising, the findings should be interpreted with caution since the subgroup analyses were not pre-specified or powered, and further investigation is needed.

In addition, considering that the subgroup analyses were just for exploration, and with small sample size, we moved the original Figure 3 into the Supplementary Fig. 1.

5. Figure 2b only shows 39 patients. An HR+ patient is missing. Why?

Response: Thanks for your comments. Target lesions in one patient were unchanged from baseline, so only a blank space was shown in the original figure. We have added a symbol “#” above the blank space in the Figure 2b and made an explanation in the Figure legend (Line 433-434), to avoid misunderstanding.

Figure 2. ...# *One patient with HR-positive disease had no change in target lesions at the end of the treatment from baseline. HR, hormone receptor.*

6. Figure 3: x axis could show 0-100% to be more aligned with actual estimates in the figure.

Response: Thanks for your comments. We have modified the tick value from ‘0-1’ to ‘0-100%’ along the x-axis in the Supplementary Fig. 1 (original Figure 3), accordingly.

7. Figure 3: symbol typo for HR less than or equal to 10%.

Response: Thanks for your comments. We have corrected the symbol from ‘~’ to ‘-’ (shown as HR 1%-10%) in the Supplementary Fig. 1 (original Figure 3).

8. Figure 4, Extended Figures 1 and 2—please provide confidence intervals for 12-mo PFS estimates.

Response: Thanks for the comments. We have added the 95% confidence intervals for 12-mo PFS% in the Supplementary Fig. 2 & Supplementary Fig. 3 (original Extended Figures 1 and 2), accordingly.

9. 12 and 18-month OS estimates are provided in the text. Please give confidence intervals for these estimates.

Response: Thanks for the comments. We have added the 95% confidence intervals for 12 and 18-month OS estimates in the Results (Line 82-83):

The estimated 12-month and 18-month OS rates were 90.0% (95% CI 75.5%-96.1%) and 82.5% (95% CI 66.8%-91.2%), respectively.

10. The authors discuss the patients with brain mets in their study. The authors should note that these patients did not have clinically symptomatic brain mets which was an exclusion criterium in their protocol. The conclusion that dalpicilib and pyrotinib may be an active regimen for HER2+ mBC with active BMs needs to be modified for transparency.

Response: Thanks for the comments. We have modified the description in the Results (Line 84) and Discussion (Line 170, 212):

*There were 13 patients with **asymptomatic** brain metastasis (BM) at baseline,...*

*Our study included 13 patients (31.7%) with clinically **asymptomatic** BM...*

...and patients with *asymptomatic* active intracranial disease could also benefit from it.

Reviewer #3 - Breast cancer clinical trials (Remarks to the Author):

The authors of this study are to be congratulated for designing and conducting a study that is supported by a sound preclinical rationale. However, the study shows weaknesses that preclude its completeness and generalizability. First, it is a single-arm study, with 32% of enrolled patients being anti-HER2 naive.

Response: Thanks for the comments. We agree with you that the single-arm design is indeed a limitation of this study, and we have stated it in the Discussion.

In addition, the patients included in the study were non-selective. Patients who had received no more than one line of prior advanced systemic treatment were included. Of 13 patients with anti-HER2 naïve, 3 were treatment-naïve, 10 were pretreated without HER2-targeted agents.

Unfortunately, there is still a significant proportion of BC patients in China who are unable to receive HER2-targeted treatment in the first-line due to the poor accessibility of pertuzumab and financial burden. In this study, 10 pretreated patients enrolled could not afford HER2-targeted agents before, and they were treated with study drugs as second-line treatment.

Second, there is no attempt to characterize responders although the study population is very small (n=40). It would be important to have information on any biomarkers that could identify which patients are most likely to benefit from a combination of CDK4/CDK6 and HER2 inhibition. Not least because, authors offer a counterintuitive finding to previous studies, as patients with HR-negative tumors appear to be the most responsive to treatment.

Response:

Thanks for your constructive advice. Unfortunately, we are unable to conduct the biomarker analysis owing to insufficient amount of tissue sample collected through puncture biopsy of advanced stage patients after various diagnostic tests needed for treatment, and limited financial resources in this study.

We have discussed this point in the limitations of this study in the revision (Line 201-204):

Secondly, biomarker analysis might provide more information to identify who can derive more benefit from this combination, but such analysis was absent due to insufficient amount of collected tumor tissue and limited financial resources.

The data on patients with brain metastasis (31% of the whole study population) is interesting given the tendency of HER2-positive breast cancer to involve the brain. However, the study was not designed or powered to investigate this issue, neither the authors comment why median PFS was overall less than one year, which is quite disappointing when we consider the current standard of care and those soon to come.

Response: Thanks for your comments. The preliminary results showed that the combination is feasible for patients with HER2-positive BC, including those with brain metastasis. However, as you point out, the subgroup analyses of brain metastasis were not pre-specified or powered. We have stated that the subgroup analyses were not pre-specified or powered, and should be interpreted with caution in the Discussion (206-209), accordingly.

Fourthly, though the efficacy of daltapiclib and pyrotinib in patients with HR-negative disease and in the ones with brain metastases was promising, the findings should be interpreted with caution since the subgroup analyses were not pre-specified or powered, and further investigation is needed.

Additionally, we agree with you that the efficacy is not satisfactory in terms of the PFS results. It may be partially because 92.7% of patients enrolled in DAP-HER-01 had visceral metastases, compared with a proportion of 67%-82% reported in other phase 3 trials, which may have affected PFS outcome. In addition, the study treatment was without endocrine therapy and estrogen receptors signaling could be an escape mechanism to bypass HER2, which may lead to the inferior efficacy in the HR-positive subtype in our study and may contribute to the suboptimal PFS outcome in the overall population.

We have add the description in the Discussion (Line 137-140, Line 151-155)

We have noted that there is a discrepancy between ORR and PFS. It may be partially because 92.7%

of patients enrolled in DAP-HER-01 had visceral metastases, compared with a proportion of 67%-82% reported in aforementioned phase 3 trials^{9, 10, 17, 18, 19}, which may have affected PFS outcome.

...

However, the inferior efficacy observed in the HR-positive subtype in our study may be attributed to the absence of inhibition on estrogen receptor signaling, which could be an escape mechanism to circumvent HER2 inhibition²¹, thus may further contribute to the suboptimal PFS outcome in the overall population.

Our future research will incorporate anti-HER2 monoclonal antibody or endocrine therapy to this combination, with the aim of enhancing its efficacy.

The discussion is verbose and authors report comparisons with existing literature without making any effort to point out the difference in study design, and population size and characteristics among the cited clinical trials.

Response: Thanks for your kind suggestion. We have made the description more concise and made extensive modifications in the Discussion. The main changes are summarized as below:

- 1) Data from previous studies of CDK4/6i combined with anti-HER2 in the treatment of HER2+ advanced breast cancer were added, and the possible reasons for the unsatisfactory efficacy in previous reports was discussed (Line 122-130).

Prior to our study, Goel S, et al.¹³ and Ciruelos E, et al.¹⁴ have assessed the use of continuous low-dose ribociclib or a three-week regimen of 200 mg palbociclib combined with trastuzumab in HER2-positive breast cancer including patients with HR-negative diseases, but both studies showed limited activity, especially in HR-negative patients. It may be due to the fact that participants in these two studies were treated with trastuzumab and heavily pretreated. In the current phase 2 study, we enrolled HER2-positive patients with no more than one line of prior advanced systemic treatment, and used dalpiciclib and pyrotinib as the therapeutic intervention. Our results showed 70% achieving an objective response, and the median PFS was 11.0 months, which were consistent with the results in the pyrotinib plus capecitabine group in PHOEBE⁹ and PHENIX¹⁰ studies.

- 2) We discussed the reasons for the short PFS in this study, pointing out that the higher proportion of patients with visceral metastasis and the inferior efficacy observed in HR+ patients in this study may have affected the overall PFS results (Line 137-140, 149-155).

We have noted that there is a discrepancy between ORR and PFS. It may be partially because 92.7% of patients enrolled in DAP-HER-01 had visceral metastases, compared with a proportion of 67%-82% reported in aforementioned phase 3 trials^{9, 10, 17, 18, 19}, which may have affected PFS outcome.

Consistently, the current study shows that HR-negative patients tend to achieve higher ORR (81.8% vs 55.6%) and longer PFS (19.3 vs 9.1 months), which is comparable with data on THP as first-line treatment in PUFFIN (ORR 75.6%, PFS 14.5 months)¹⁵. However, the inferior efficacy observed in the HR-positive subtype in our study may be attributed to the absence of inhibition on estrogen receptor signaling, which could be an escape mechanism to circumvent HER2 inhibition²¹, thus may further contribute to the suboptimal PFS outcome in the overall population.

Reviewers' Comments:

Reviewer #1:

Remarks to the Author:

All comments have been properly addressed. The only thing I would change is the part of MAPK. The comment was done to underline the importance of biomarker analysis in this type of studies, rather than making a point specifically on MAPK pathway alterations. I would remove or change in a more generic way the following sentence: "Thus, analyzing the alteration of MAPK pathway and other signaling related to tumor growth and progression during the treatment is needed in future studies."

Reviewer #2:

Remarks to the Author:

Thank you for your responses. My prior comments have all been adequately addressed. I have two new comments arising from the corrections/edits:

1. The estimates and confidence intervals in the Efficacy section do not always align with Figure 3.
2. The confidence intervals in the Efficacy section do not always align with Supplementary Figure 3.

Reviewer #3:

Remarks to the Author:

I appreciate the authors' attempt to improve their manuscript. However, I would be lying if I said that their responses convinced me. I believe there is a fundamental flaw, namely, that the authors present a study that could have been an opportunity for safety evaluation, tumor activity, and drug mechanisms of action, as a definitive one. For example, they talk about efficacy rather than activity. This approach, presenting a small, single-arm Phase II study (with some calculation errors that have been corrected in this version) as if it were a registrative trial, leads the reader, at least myself, to immediately think that the study is flawed in its design, endpoint, patient population, and lack of correlative science.

Moreover, even after reviewing the new version, I am doubtful that the results can truly be considered generalizable because the study population consisted of untreated patients with brain metastases (which is somewhat counterintuitive). It needs to be contextualized within the authors' reality. The beauty of the study lies in its focus on two chemofree drugs, but the results obtained don't excite me because we are accustomed to stronger chemofree combinations. Additionally, we are intrigued by the possible mechanism of action of the combination in the negative receptors. The authors should take inspiration from the feedback of Reviewer#1 to highlight this point.

It would be beneficial if the work were to be rewritten, downsized, and the missing parts (involving 40 patients) were completed to create a provocative and informative report. Unfortunately, in my opinion, the study does not deserve to be published in its current form; it requires a complete revision.

Point-by-Point responses

REVIEWERS' COMMENTS

Reviewer #1 (Remarks to the Author):

All comments have been properly addressed. The only thing I would change is the part of MAPK. The comment was done to underline the importance of biomarker analysis in this type of studies, rather than making a point specifically on MAPK pathway alterations. I would remove or change in a more generic way the following sentence: "Thus, analyzing the alteration of MAPK pathway and other signaling related to tumor growth and progression during the treatment is needed in future studies."

Response: Thanks for your positive comments and helpful suggestions. We have revised the sentence "Thus, analyzing the alteration of MAPK pathway and other signaling related to tumor growth and progression during the treatment is needed in future studies." (Line 151-152) into "Thus, analyzing the potential mechanisms underlying tumor growth and progression during the treatment is needed in future studies."

Reviewer #2 (Remarks to the Author):

Thank you for your responses. My prior comments have all been adequately addressed. I have two new comments arising from the corrections/edits:

1. The estimates and confidence intervals in the Efficacy section do not always align with Figure 3.

Response: Thanks for your suggestions. We are so sorry for our negligence. We have checked the data carefully, and confirmed that the estimated 12-month progression-free survival was 44.7% not 44.8%. Data in the main text has been revised. (Line 72)

In addition, the median PFS with 95% CI data of HR-negative patients has been revised from 19.3 (7.7-30.1) to 19.3 (7.4-30.1) in the Fig 3.

2. The confidence intervals in the Efficacy section do not always align with Supplementary Figure 3.

Response: We apologize for our mistakes. We have checked the data again and the 95% confidence interval of the median PFS among all BM patients is 5.5-33.9. Data in the main text has been revised. (Line 116)

Reviewer #3 (Remarks to the Author):

I appreciate the authors' attempt to improve their manuscript. However, I would

be lying if I said that their responses convinced me. I believe there is a **fundamental flaw**, namely, that the authors present a study that could have been an opportunity for safety evaluation, tumor activity, and drug mechanisms of action, as a definitive one. For example, they talk about efficacy rather than activity. **This approach, presenting a small, single-arm Phase II study (with some calculation errors that have been corrected in this version) as if it were a registrative trial**, leads the reader, at least myself, to immediately think that the study is flawed in its design, endpoint, patient population, and lack of correlative science.

Moreover, even after reviewing the new version, I am doubtful that the results can truly be **considered generalizable** because the study population consisted of **untreated patients with brain metastases** (which is somewhat counterintuitive). It needs to be contextualized within the authors' reality. The beauty of the study lies in its focus on two chemofree drugs, but **the results obtained don't excite me** because we are accustomed to stronger chemofree combinations. Additionally, we are intrigued by the **possible mechanism of action** of the combination in the negative receptors. The authors should take inspiration from the feedback of Reviewer#1 to highlight this point.

It would be beneficial if the work were to be rewritten, downsized, and the missing parts (involving 40 patients) were completed to create a provocative and informative report. Unfortunately, in my opinion, the study does not deserve to be published in its current form; it requires a complete revision.

Response: Thank you for this valuable feedback. Your suggestions truly reflect the limitations of this study, as well as allow us to have a more in-depth view of our work. In order to let readers view this research more objectively, we have listed the limitations you mentioned above in the Discussion. We hope our study may inspire more insight for the treatment of HER2-positive BC. In addition, we have changed "efficacy" to "activity" throughout the manuscript. Thanks again.